# Impacts of MicroRNA-483 on Human Diseases

**DOI:** 10.3390/ncrna9040037

**Published:** 2023-06-28

**Authors:** Katy Matson, Aaron Macleod, Nirali Mehta, Ellie Sempek, Xiaoqing Tang

**Affiliations:** Department of Biological Sciences, Michigan Technological University, Houghton, MI 49931, USA; matson@mtu.edu (K.M.); abmacleo@mtu.edu (A.M.); nmehta2@mtu.edu (N.M.); essempek@mtu.edu (E.S.)

**Keywords:** miRNAs, miR-483-5p, miR-483-3p, diabetes, biomarker

## Abstract

MicroRNAs (miRNAs) are short non-coding RNA molecules that regulate gene expression by targeting specific messenger RNAs (mRNAs) in distinct cell types. This review provides a com-prehensive overview of the current understanding regarding the involvement of miR-483-5p and miR-483-3p in various physiological and pathological processes. Downregulation of miR-483-5p has been linked to numerous diseases, including type 2 diabetes, fatty liver disease, diabetic nephropathy, and neurological injury. Accumulating evidence indicates that miR-483-5p plays a crucial protective role in preserving cell function and viability by targeting specific transcripts. Notably, elevated levels of miR-483-5p in the bloodstream strongly correlate with metabolic risk factors and serve as promising diagnostic markers. Consequently, miR-483-5p represents an appealing biomarker for predicting the risk of developing diabetes and cardiovascular diseases and holds potential as a therapeutic target for intervention strategies. Conversely, miR-483-3p exhibits significant upregulation in diabetes and cardiovascular diseases and has been shown to induce cellular apoptosis and lipotoxicity across various cell types. However, some discrepancies regarding its precise function have been reported, underscoring the need for further investigation in this area.

## 1. Introduction

Type 2 diabetes (T2D) is a chronic metabolic disorder characterized by elevated blood glucose levels due to deficiencies in insulin secretion by pancreatic β-cells and/or impaired insulin sensitivity in peripheral tissues such as the liver, adipose tissue, and muscle [1,2]. Individuals with T2D face an augmented risk of developing severe secondary complications, including fatty liver disease, endothelial dysfunction, cardiovascular disease, and renal failure [3].

MicroRNAs (miRNAs) are a class of endogenous non-coding RNA approximately 17–23 nucleotides in length that bind to the 3′-untranslated region (3′UTR) of target mRNAs, leading to their degradation and/or translational repression [4]. Dysregulation of miRNA expression has been linked to a variety of diseases, including diabetes, making them potential biomarkers and therapeutic targets [5,6]. MiRNAs exhibit dynamic regulation of gene expression, which facilitates buffering of gene expression levels to achieve a stable state. However, miRNA-mediated inhibition of target transcripts is not universal among cell types, as it can be influenced by alternative splicing/polyadenylation events and the levels of cell type-specific factors that can alter the secondary structure of target transcripts [7].

MicroRNAs have the ability to circulate in a stable manner in the blood, either in association with protein complexes or encapsulated within exosome-like vesicles. They play a crucial role in facilitating communication between different cell types [8]. The accessibility of circulating miRNAs makes them promising biomarkers, as they can be detected in various bodily fluids, including blood plasma, serum, urine, cerebrospinal fluid, and saliva. To ensure their protection and transportation, miRNAs are enveloped by lipids, exosomes, or proteins such as Argonaute2 (AGO2) [9]. These circulating miRNAs may act as mediators of endocrine signaling between organs or serve as indicators of tissue-specific miRNA release through paracrine or autocrine signaling within an organ.

The dysregulation of miRNA expression is not the sole determinant of human disease development, yet it does contribute to the regulation of disease progression and prognosis. Consequently, rectifying the imbalance of miRNA expression levels holds significant therapeutic potential. Disease-modifying therapies that target miRNAs can be devised through two primary strategies: increasing protective mRNA levels by inhibiting miRNAs or decreasing detrimental mRNA through miRNA mimics [10]. Recent advancements in oligonucleotide delivery technology have demonstrated the ability to selectively target specific tissues, cells, and subcellular compartments, including the traversal of the blood–brain barrier for brain-specific delivery [10]. These breakthroughs in RNA therapeutics have instilled optimism that these approaches may eventually translate into viable treatment options in the future.

This review provides a comprehensive summary of the existing understanding regarding the tissue-specific expression patterns of miR-483 in various organs and underscores its diverse functions in numerous diseases including obesity, diabetes, fatty liver disease, heart failure, diabetic nephropathy, and brain injury.

## 2. MiR-483 Biogenesis

MiR-483 is located within the second intron of the human insulin-like growth factor 2 (*IGF2*) gene on chromosome 11 (Figure 1). It encodes for two mature microRNAs: miR-483-5p and miR-483-3p. The biogenesis of miR-483 follows the standard miRNA biogenesis process, involving transcription of the primary miRNA transcript (pri-miR-483), which is then cleaved by the nuclear microprocessor complex to produce a precursor miRNA (pre-miR-483). After export from the nucleus, the stem-loop pre-miR-483 is further processed by Dicer to produce double-stranded duplexes, which include miR-483-5p derived from the 5′-arm (previously called miR-483), and miR-483-3p derived from the 3′-arm (previously called miR-483*). Traditionally, it was believed that only one of the miRNA strands is stably produced and functional, while the other strand is degraded. However, more recent studies have shown that both mature miRNA strands can be expressed and have functional roles for most miRNAs. MiR-483-5p and miR-483-3p are both produced and functional in specific tissues and cells. They exhibit partial reverse complementarity and have distinct mRNA-targeting sequences, which makes them biologically different and capable of targeting different mRNA transcripts (Figure 1). The sequences of miR-483-5p and miR-483-3p are highly conserved among mammalian species, including humans, mice, and rats.

The genomic localization of miR-483 is intriguing, as it resides within the *INS-IGF2* locus, which plays a critical role in the insulin pathway [11]. The host gene for miR-483, *IGF2*, is located within the *IGF2/H19* genetic locus [12]. Notably, *IGF2* is exclusively expressed from the paternal allele, while the *H19* gene demonstrates maternal expression. IGF2 has established involvement in lipid metabolism and obesity [13].

Both miR-483-5p and miR-483-3p have demonstrated functional activity, although their differences in terms of transcription regulation and stability remain poorly understood. Studies have suggested that certain intragenic miRNAs are co-expressed and function in conjunction with their host genes, while others do not exhibit such coordination. The transcription of miR-483-5p may be coregulated with its host gene, *IGF2*, or regulated independently of *IGF2* transcription [14]. Considering that *IGF2* transcript levels are typically very low in adults, it is expected that *IGF2* and miR-483 are independently regulated in adult tissues. Aberrant expressions of miR-483 regulated by DNA methylation have been observed in the liver and skeletal muscle [15]. In addition, miR-483-5p is reported to be upregulated by high glucose, insulin, and various stress conditions [16,17,18], suggesting the expression of miR-483-5p is regulated in response to changes in cellular environment in order to support adaptation to an altered microenvironment. The transcription of miR-483-3p can also be upregulated independently of *IGF2* transcription via β-catenin/Wnt signaling [19], insulin-responsive/sensitive elements, or inflammatory transcription factors [20]. These data suggest that the expression patterns of miR-483-5p and miR-483-3p may vary across different cell types and can even differ within the same cells under various pathological stress conditions. 

## 3. Role of MiR-483-5p in Diabetes

### 3.1. MiR-483-5p Protects Pancreatic β-Cells Function and Identity

Pancreatic β-cells are specialized cells in the pancreas that synthesize and secrete insulin, a hormone that regulates blood glucose level. In patients with T2D, β-cells gradually lose their function and secrete insufficient insulin, which leads to elevated blood glucose levels [21]. A common view is that the continuous loss of β-cells is a major feature in the development of type 2 diabetes. However, this principle has been challenged by studies suggesting that β-cells do not die in patients with diabetes but undergo alteration in β-cell identity (or β-cell dedifferentiation) [22,23]. The dedifferentiation of β-cells into glucagon-producing α-cells or the earlier-stage progenitor-like cells has been considered a feature during the development of T2D.

MiR-483-5p has been shown to exhibit highly differential expression between pancreatic β-cells and α-cells, with higher expression in β-cells than in α-cells [16]. This differential expression is critical for maintaining β-cell function since miR-483-5p targets the suppressor of cytokine signalling3 (*SOCS3*), leading to increased insulin transcription in β-cells and decreased glucagon transcription in α-cells (Table 1, Figure 2). Mice with β-cell-specific deletion of miR-483-5p display hyperglycemia and glucose intolerance, along with reduced insulin release when fed a high-fat diet [24].

Deletion of miR-483 results in the loss of β-cell identity, as evidenced by elevated expression of aldehyde dehydrogenase family 1, subfamily A3 (*ALDH1A3*) [24]. ALDH1A3 has been validated as a novel marker of β-cell dedifferentiation, and elevated ALDH1A3 expression occurs in the islets of diabetic human subjects and various diabetic mouse models [25]. Further studies have validated ALDH1A3 as a direct target of miR-483-5p, and overexpression of miR-483-5p represses ALDH1A3 expression [24] (Table 1, Figure 2). However, the identity of ALDH1A3-positive β-cells and how miR-483-5p participates in ALDH1A3-mediated β-cell dedifferentiation remains to be investigated.

Hyperlipidemia, characterized by reduced circulating high-density lipoprotein (HDL) cholesterol and elevated circulating low-density lipoprotein (LDL) cholesterol, is a major risk factor for developing diabetes [26]. Genetic ablation of miR-483 induces unfavorable blood lipid profiles, including increased LDL and decreased HDL [24]. High serum levels of cholesterol are associated with increased islet cholesterol content and decreased insulin secretion [26]. Islets exposed to LDL show an increase in oxidative stress, leading to impaired mitochondrial function and insulin synthesis. These findings highlight the critical role of miR-483-5p in protecting β-cell function, and miR-483-5p inactivation induces dyslipidemia and initiates β-cell dedifferentiation.

Exosomes from human islets are enriched with miR-483-5p more than 10-fold compared to cellular content [27]. Furthermore, the level of miR-483-5p in serum is strongly associated with metabolic risk factors for diabetes, such as body mass index (BMI), waist circumference, insulin resistance, triglycerides, and cholesterol levels [18]. The circulating miR-483 levels are also coordinated with the change in glycated hemoglobin A1C (HbA1c) in T2D patients with short-term intensive insulin therapy [28]. These findings suggest that miR-483-5p holds immense potential as a novel diagnostic and prognostic tool in T2D treatment strategies.

### 3.2. MiR-483-5p Mitigates Hyperlipidemia-Associated Fatty Liver Disease

The liver plays a central role in lipid metabolism by secreting LDL via the LDL receptor (LDLR) on the liver surface [29]. Excessive hepatic lipid accumulation can trigger cellular stress responses, inflammation, cell death, and fibrosis [30]. Thus, hyperlipidemia in individuals with insulin resistance and T2D is a significant factor in the development of hepatic insulin resistance and the progression of non-alcoholic fatty liver disease (NAFLD) or alcoholic fatty liver disease (AFLD) [31].

Research has shown that hyperlipidemic human subjects with elevated LDL levels exhibit lower levels of circulatory miR-483-5p [32]. In hypercholesterolemic mouse models and HepG2 cells, overexpression of miR-483-5p significantly inhibits the expression of *PCSK9*, which subsequently increases hepatocyte expression of LDLR and enhances LDL uptake [33] (Figure 2). *PCSK9* encodes the proprotein convertase subtilisin/kexin type 9 protein, which binds LDL and induces LDLR degradation. *PCSK9* loss-of-function mutations are associated with hypocholesterolemia, while gain-of-function mutations are linked to familial hypercholesterolemia [27,34]. PCSK9 has been identified as a promising therapeutic target for patients who do not adequately respond to statins, a commonly used cholesterol-lowering medication [35]. Further studies have confirmed that PCSK9 is a direct target of miR-483-5p, and overexpression of miR-483-5p has also been shown to reduce plasma cholesterol and LDL levels in hypercholesterolemic mice (Table 1). These findings suggest that miR-483-5p can ameliorate hypercholesterolemia and may be a potential therapeutic target for this condition.

A previous study has found that overexpression of miR-483-5p inhibits the expression of *SOCS3* in mouse Hepa1-6 cells [14] (Table 1). SOCS3 is a negative regulator of the JAK/STAT pathway and is implicated in hypertriglyceridemia associated with insulin resistance and leptin resistance [36,37]. New evidence shows that miR-483 is downregulated in rats with liver fibrosis [38] (Table 1, Figure 2). Overexpression of miR-483 inhibits mouse liver fibrosis by targeting tissue inhibitor of metalloproteinase 2 (*TIMP2*) and platelet-derived growth factor-β (*PDGF-β*). A recent report indicates that miR-483-5p expression is lower in NAFLD and AFLD mouse models and human hepatocellular carcinoma (HCC) tissue samples [39]. The downregulation of miR-483-5p increases not only *TIMP2* expression but also peroxisome proliferator-activated receptor alpha (*PPARα*) and transforming growth factor beta (*TGF-β*) expression (Table 1, Figure 2). In the liver, PPARα regulates lipid metabolism and controls liver homeostasis, and its dysregulation may lead to hepatic steatosis and fibrosis [40]. These findings suggest that miR-483-5p can alleviate lipid deposition in the liver and mitigate hyperlipidemia-associated NAFLD. Therefore, miR-483 may be a potential therapeutic target for treating patients with fatty liver disease.
ncrna-09-00037-t001_Table 1Table 1Impacts of miR-483-5p in human diseases.Type of DiseaseExpression PatternTissueCell LinesTargetsFunctionReferencesType 2 Diabetesdownpancreatic isletsMIN6*SOCS3**ALDH1A3*Induce insulin secretion, inhibit glucagon secretion, and maintain β-cell identity[16,24]Obesity/Diabetesdownadipose3T3-L1*ERK1**MeCP2*Promote adipogenesis[41,42,43]Fatty liver disease (NAFLD/AFLD)downliverHepG2*PCSK9**TIMP2**TGF-β**PPARα**SOCS3*Reduce lipid deposition and inhibit liver fibrosis[32,33,38,39]Diabetic nephropathydownKidney tubuleHK2TCMK-1*HDAC4**TIMP2**MAPK1*Prevent renal tubular damage and renal fibrosis[44,45]Alzheimer,Brain injury after cardiac arrestdownneuronNeonatal Fibroblasts,PC12*ERK1/2**TNFSF8**MeCP2*Promote mitochondrial biogenesis, inhibit ROS generation, protect neurological function,and regulate fetal brain development[46]Cardiovascular diseaseupserum,carotid bulbAC16*MAPK3*Induced cell apoptosis and oxidative stress[18,47,48,49]

### 3.3. MiR-483-5p Promotes Adipogenesis in the Adipose

Adipose tissue plays essential roles in maintaining lipid and glucose homeostasis. However, in obesity, adipose tissue becomes dysfunctional, promoting a pro-inflammatory, hyperlipidemic, and insulin-resistant environment that contributes to T2D and cardiovascular disease [50]. Adipogenesis is a complex series of steps involving the orchestrated activation of multiple transcription factors, which drive the typical physiological and morphological changes [51]. Among these factors, the expression of peroxisome proliferator-activated receptor gamma (PPARγ) plays a crucial role in facilitating the differentiation and maintenance of mature adipocytes. PPARγ induces the expression of genes associated with insulin sensitivity, lipogenesis, and lipolysis, thereby influencing the metabolic functions of adipocytes [52].

Recent studies have shown that miR-483-5p plays a significant role in regulating adipogenesis. The expression of miR-483-5p is significantly downregulated in subcutaneous adipose tissue obtained from human subjects with obesity compared to those without obesity [41] (Table 1). MiR-483-5p positively regulates the expression of *PPARγ* and facilitates adipogenesis in mouse pre-adipocyte 3T3-L1 cells and human adipose-derived MSCs by targeting *ERK1* [42,53]. It also targets methyl CpG-binding protein 2 (*MeCP2*), an epigenetic negative regulator of adipose browning, and enhances adipogenesis in hBMSCs [43,54] (Table 1, Figure 2). Inhibition of miR-483-5p fails to stimulate the accumulation of marrow adipocytes and PPARγ expression.

Moreover, miR-483-5p can influence osteogenesis, a bone disease characterized by low bone mass and poor bone quality. It is highly enriched in extracellular vesicles derived from aged bone matrix, and its inhibition delays the development of osteoarthritis [55,56]. Additionally, miR-483-5p can be released from differentiating adipocytes and found in the plasma of subjects with osteoporosis, indicating its potential as a non-invasive biomarker for this condition.

## 4. Role of MiR-483-5p in Other Human Diseases

Hyperglycemia associated with diabetes can damage various organs, including the cardiovascular system and kidneys, over time. Diabetic complications include nephropathy, neuropathy, cardiomyopathy, and retinopathy [57]. They share similar pathological mechanisms, such as inflammation, mitochondrial dysfunction, and oxidative stress. Recent studies have demonstrated that miR-483-5p participates in the pathogenesis of diabetic complications.

### 4.1. MiR-483-5p Downregulation in Patients with Diabetic Nephropathy

Diabetic nephropathy (DN) is a common microvascular complication of diabetes and is characterized by progressive kidney disease. The disease initially manifests as glomerular hyperfiltration, followed by persistent albuminuria and microalbuminuria, ultimately leading to significant proteinuria, impaired renal function, and a gradual reduction in the glomerular filtration rate [58].

Recent studies have demonstrated that miR-483-5p is downregulated in kidney tissues of both type 1 and type 2 diabetic mouse models, as well as in proximal renal tubular cells of patients with DN [44] (Table 1, Figure 2). The level of miR-483-5p is positively correlated with estimated glomerular filtration rate (eGFR), an indicator of renal function. MiR-483-5p reduction is associated with lower eGFR and prevalence of proteinuria. Overexpression of miR-483-5p has been shown to have a protective effect on high glucose-induced damage in renal tubular cells by targeting *HDAC4* [44]. Furthermore, miR-483-5p targets *TIMP2* and *MAPK1* in renal tubular epithelial cells [45], both of which are involved in the pathogenesis of renal disease (Table 1, Figure 2). Notably, the expression of *TIMP2* is significantly higher in renal biopsies with renal disease [59]. Injection of AAV-miR-483-5p has been shown to alleviate interstitial fibrosis in diabetic mice by inhibiting the expression of *TIMP2, MAPK1*, and other fibrosis-related genes such as *Col1a1, Col4a1*, and fibronectin [59].

Interestingly, miR-483-5p is present in urine and is upregulated in the extracted urine exosomes of DN patients [45]. MiR-483-5p can be secreted from cellular tubular epithelial cells and transported into urine through exosomes. The process is mediated by hnRNPA1, a key player in miRNA sorting and transporting into extracellular vesicles [60]. Differences in miR-483-5p expression in tubular epithelial cells between intracellular and extracellular environments may provide new insights for the diagnosis and treatment of DN. The accumulation of miR-483-5p in urinary exosomes may serve as a new biomarker for effectively distinguishing DN patients from healthy individuals. 

### 4.2. MiR-483-5p Protects Neurological Function against Oxidative Stress

According to recent studies, miR-483-5p plays a crucial role in protecting against neurological injury induced by cardiac arrest. Specifically, miR-483-5p is significantly downregulated in hippocampal samples from patients with neurological injury after cardiac arrest compared to the normal group [61] (Table 1, Figure 2). Brain injury following cardiac arrest is mainly caused by the generation of mitochondrial reactive oxygen species (ROS), which can be inhibited by promoting mitochondrial biogenesis and reducing ROS generation. MiR-483-5p targets *TNFSF8* to achieve this effect and protect against neurological impairment. Overexpression of miR-483-5p reduces protein expression of Bax and cleaved caspase 3, thereby inhibiting cytochrome c release and ROS generation from mitochondria, which alleviates cell injury by ischemia–reperfusion [61]. The inhibition of miR-483-5p in rats results in more severe hippocampal damage than the control group after cardiac arrest. These findings demonstrate that miR-483-5p targets TNFSF8 to protect against neurological impairment by promoting mitochondrial biogenesis and inhibiting ROS generation.

Previous research also suggests that miR-483-5p elevation represents a compensatory neuroprotective mechanism during the early stage of Alzheimer’s disease [62]. During the early stage of Alzheimer’s disease, miR-483-5p upregulation inhibits TAU phosphorylation and prevents neurofibrillary pathology by targeting *ERK1/2* expression (Table 1, Figure 2). In contrast, the depletion of miR-483-5p results in a concomitant increase in pathological TAU phosphorylation and neurofibrillary tangle formation in the brain. MiR-483-5p is co-upregulated with its host gene *IGF2* in the cerebrospinal fluid of patients with Alzheimer’s disease. IGF2 upregulation can promote neurogenesis and synapse formation against oxidative stress, suggesting a potential neuroprotective role for miR-483-5p in Alzheimer’s disease.

Furthermore, miR-483-5p plays a crucial role in the regulation of MeCP2 levels for brain development during fetal stages [46] (Table 1). MeCP2 protein levels must be tightly regulated to ensure normal neurological function, and miR-483-5p precisely controls MeCP2 levels during fetal development. MeCP2 levels are repressed during fetal development and elevated during postnatal development in human brains. MiR-483-5p is enriched in human fetal brains compared to postnatal brains, indicating an inverse correlation between miR-483-5p and MeCP2 levels in developing human brains. In addition, overexpression of miR-483-5p in hippocampal neurons rescues the abnormal dendritic spine phenotype caused by elevated MeCP2. MiR-483-5p modulates MeCP2-interacting corepressor complexes, including HDAC4 and TBL1X, shedding light on its role in regulating MeCP2 levels and interacting proteins during human fetal development.

### 4.3. Elevation of Circulating MiR-483-5p as a Biomarker for Cardiovascular Disease

Cardiovascular disease (CVD) is the primary cause of mortality and morbidity, and atherosclerosis serves as the prevailing underlying factor in CVD-related manifestations, including heart attacks and strokes. Individuals with diabetes have historically exhibited a higher prevalence of CVD compared to those without diabetes, and the risk of CVD often coincides with comorbidities such as obesity, dyslipidemia, and hypertension [63]. 

Studies have shown that elevated serum levels of miR-483-5p are significantly associated with insulin resistance and hypertension, which are risk factors for the development of T2D, cardiovascular disease, and other associated complications [18,64] (Table 1). The serum concentration of miR-483-5p could reflect the degree of leakage from the source tissue due to varying degrees of permeability/leakage. In fact, miRNAs can be secreted as exosomes to transfer their effects to adjacent cells [9]. The increase in miR-483-5p in serum could be a consequence of tissue damage or inflammation during the development of diabetes and cardiovascular disease.

Indeed, elevated circulating levels of miR-483-5p are observed in patients with acute myocardial infarction (AMI) compared to those without AMI [47] (Table 1, Figure 2). AMI, commonly known as a heart attack, is a serious condition that occurs when a blood clot obstructs blood flow to the heart, leading to cardiac cell death and impaired heart function. Timely diagnosis is crucial for effective management and treatment. Research has shown that circulating miR-483-5p levels positively correlate with elevated cardiac troponin I, a widely used biomarker for AMI diagnosis [48]. In fact, miR-483-5p reaches peak expression much earlier than troponin I, suggesting miR-483-5p would provide more sensitive and valuable diagnostic information than traditional markers during the early phases of AMI.

Despite the observed rise in circulating levels of miR-483-5p, studies also showed a predominant increase in miR-483-5p expression in the atrial myocardium of patients diagnosed with atrial fibrillation (AF) compared to individuals maintaining sinus rhythm [49] (Figure 2). AF is a common type of cardiac arrhythmia characterized by irregular electrical activities in the atria. It often coexists with AMI. In vitro experiments demonstrated that hypoxia induces miR-483-5p expression in cardiomyocytes and miR-483-5p upregulation increases hypoxia-induced cell apoptosis and oxidative stress by targeting *MAPK3* [48] (Table 1). Furthermore, miR-483-5p has been found to be significantly linked to intima media thickness of the carotid bulb and the number of plaques, both of which are indicators of atherosclerosis [65,66]. The available data collectively indicate that miR-483-5p holds promise as a promising biomarker for assessing the risk of CVD. However, the causal relationship between the increased levels of miR-483-5p and the cardiac rhythm disorder remains uncertain. Further investigations are necessary to elucidate the underlying mechanisms and fully understand the role of miR-483-5p in the pathogenesis of diabetes and CVD.

## 5. Implication of MiR-483-3p in Human Disease

MiR-483-3p has been extensively studied in the context of cancer [67]. More recent data have shown that mir-483-3p upregulation has been reported in adipose, cardiomyocytes, and endothelial cells in patients with diabetes or cardiovascular diseases. Its role can differ depending on the disease context or cell type, where it can act as a factor inducing apoptosis or supporting cell survival against inflammation.

In contrast to miR-483-5p, miR-483-3p has been demonstrated to inhibit adipocyte differentiation and promote insulin resistance [68] (Table 2). Studies conducted on adipose tissue from both adult humans with low-birth-weight and prediabetic adult rats exposed to suboptimal nutrition early in life have revealed a notable increase in miR-483-3p expression, suggesting a conserved programming of this miRNA between species [69] (Table 2). Overexpression of miR-483-3p hampers adipocyte differentiation by suppressing growth/differentiation factor-3 (*GDF3*), a member of the TGFβ superfamily. This leads to a reduction in the number of lipid droplet-containing cells, resulting in limited lipid storage capacity within the adipose tissue. Consequently, this dysregulation promotes lipotoxicity and insulin resistance, and it heightens vulnerability to metabolic diseases [69].

Endothelial cell injury or dysfunction is a pivotal initial event in the pathogenesis of vascular complications associated with diabetes. The expression of miR-483-3p is elevated in the aortic endothelial wall of individuals with diabetes compared to control subjects without diabetes, and it is also higher in diabetic mice compared to nondiabetic mice [20] (Table 2). Overexpression of miR-483-3p induces increased apoptosis in both endothelial cells and macrophages, impairing reendothelialization processes. This effect is mediated through the targeting of vascular endothelial zinc finger 1 (*VEZF1*), an endothelial transcription factor that is diminished in the aorta of diabetic mice [20]. In a murine carotid-injury model, systemic inhibition of miR-483-3p leads to increased VEZF1 expression and rescues the impaired reendothelialization associated with diabetes, highlighting that heightened miR-483-3p levels limit endothelial repair capacity in patients with T2D. Furthermore, in endothelial progenitor cells (EPCs), miR-483-3p targets serum response factor (*SRF*) resulting in reduced cell migration and tube formation while promoting cell apoptosis [70] (Table 2).

The upregulation of miR-483-3p serum levels has also been observed in newly diagnosed diabetic cats, streptozotocin-induced diabetic mice, or high glucose-cultured cardiomyocytes [71,72] (Table 2). Upregulated miR-483-3p in transgenic mice with diabetes exacerbates cardiomyocyte apoptosis by transcriptionally repressing insulin growth factor 1 *(IGF1*) [72]. Taken together, the high-glucose-induced upregulation of miR-483-3p induces cell apoptosis and promotes cardiomyocytes and endothelial cell injury, which contributes to the high cardiovascular risk in patients with type 2 diabetes.

However, new studies suggest that miR-483-3p may play a protective effect against endothelial dysfunction in hypertension [73]. Hypertension is a significant risk factor for the development of coronary heart disease and stroke. Endothelial dysfunction and arterial remodeling contribute to increased vascular wall thickness and arterial stiffness. Analyses of patient cohorts have revealed a correlation between serum levels of miR-483-3p and the progression of hypertension [73] (Table 2). The overexpression of miR-483-3p in endothelial cells inhibits Ang II-induced endothelial dysfunction by targeting several molecules, such as transforming growth factor-beta (*TGF-β*), connective tissue growth factor (*CTGF*), angiotensin-converting enzyme 1 (*ACE1*), and endothelin-1 (*ET-1*) [73] (Table 2). Telmisartan, a drug used to treat hypertension, can also significantly induce the aortic and serum levels of miR-483-3p in hypertension patients and spontaneous hypertension rats. 

MiR-483-3p also exerts a protective role against endothelial inflammation and prevents aortic valve (AV) calcification, a major cause of cardiac-related deaths in aging individuals [74]. The expression of miR-483-3p is reduced in AV endothelial cells during the process of porcine AV calcification, and miR-483-3p mimic protects against endothelial inflammation and inhibits AV calcification [74] (Table 2). In contrast, miR-483-3p reduction induces endothelial inflammation and subsequent AV calcification by increasing the expression of its target, ubiquitin E2 ligase-C (*UBE2C*). Moreover, a very recent report showed a similar downregulation of miR-483-3p in the cardiac tissue of mice with acute myocardial infarction (AMI) [75] (Table 2). Therapeutic hypothermia, a clinical therapy to reduce body temperature and suppress tissue injury from AMI, induces miR-483-3p expression, and elevated miR-483-3p inhibits hypoxia-induced cardiomyocyte apoptosis by directly binding and decreasing cyclin-dependent kinase 9 (*CDK9*). Together, these data suggest that miR-483-3p exerts a protective effect, and miR-483-3p mimic may serve as a potential therapeutic for cardiovascular disease. The discrepancies regarding the precise function of miR-483-3p in cardiomyocytes and endothelial cells require further investigation.

## 6. Conclusions and Perspectives

In summary, miR-483-5p and miR-483-3p exhibit distinct and tissue-specific functions in the pathogenesis of human diseases. The available evidence highlights the protective role of miR-483-5p against lipotoxicity and oxidative stress through its regulation of specific targets in various cell types. The reduced expression of miR-483-5p is associated with conditions such as type 2 diabetes, fatty liver, diabetic nephropathy, and neurological injuries. Conversely, elevated levels of miR-483-5p in serum are strongly correlated with obesity and hypertension, underscoring its potential as a valuable biomarker for the prediction and diagnosis of diabetes and CVS.

Conversely, miR-483-3p displays notable upregulation in the context of diabetes and CVS, and its involvement has been demonstrated in inducing cell apoptosis and lipotoxicity across diverse cell types. However, recent investigations have suggested a potential protective role of miR-483-3p against endothelial inflammation, highlighting the intricate nature of its function and emphasizing the need for additional research to fully comprehend its mechanisms.

Considering the expression and functional roles of both miR-483-5p and miR-483-3p in specific reported tissues and cells, further studies are needed to investigate their co-expression patterns and regulatory mechanisms. While they hold promise as biomarkers and potential therapeutic targets, it is crucial to elucidate the underlying mechanisms that contribute to their altered expression and circulation. Further research is necessary to advance our understanding in this regard.

## Figures and Tables

**Figure 1 ncrna-09-00037-f001:**
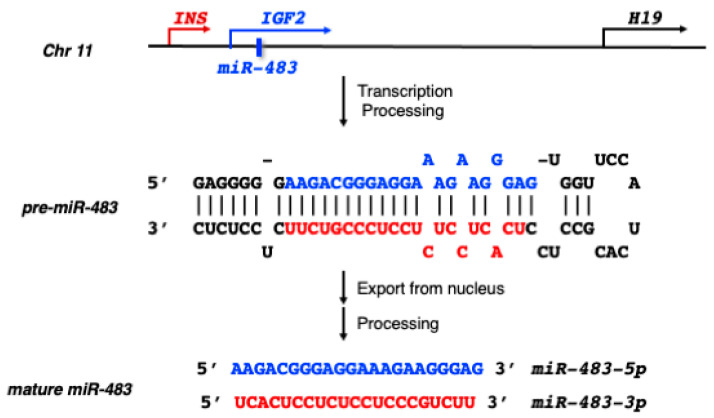
The genomic location of miR-483 and Ins-Igf2-H19 gene clusters on human chromosome 11. The sequences of the stem-loop precursor miRNA (pre-miR-483) and two mature miRNAs, miR-483-5p (highlighted in blue) and miR-483-3p (highlighted in red) are from the miRBase database (https://www.mirbase.org/, accessed on 15 May 2023).

**Figure 2 ncrna-09-00037-f002:**
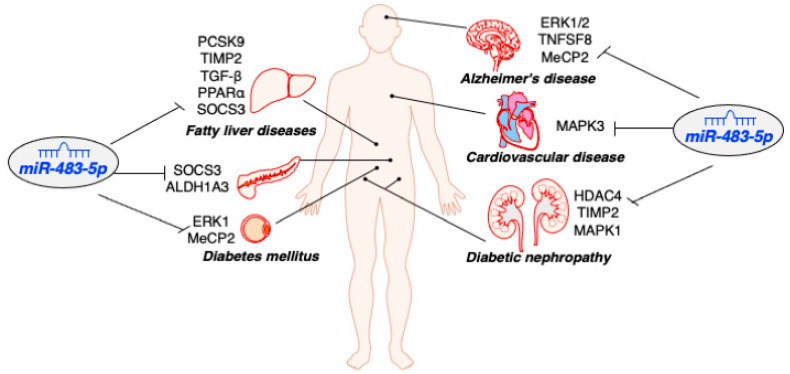
Impacts of miR-483-5p in human diseases. The cartoon highlights the involvement of miR-483-5p in numerous human diseases through its regulation of specific targets in various cell types, including liver, pancreas, adipose, kidney, heart, and brain.

**Table 2 ncrna-09-00037-t002:** Implications of miR-483-3p in human diseases.

Type of Disease	Expression Pattern	Tissue	Cell Lines	Targets	Function	Refs
prediabetes/ type 2 diabetes	up	adipose	3T3-L1	*GDF3*	Induce lipotoxicity and insulin resistance	[69]
diabetic vascular disease	up	vascular endothelium, endothelial progenitor cells, cardiomyocytes	HAEC, H9C2	*VEZF1* *SRF1* *IGF1*	Induce apoptosis in cardiomyocytes and endothelial cells	[20,70,71,72]
cardiovascular disease, hypertension, aortic valve calcification	down	serum, heart/aortic valve endothelial cells, cardiomyocytes		*TGF-β* *CTGF* *ACE1* *ET1* *UBE2C* *CDK9*	Inhibit apoptosis, Protects endothelial function against inflammation	[73,74,75]

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
