# Peer review of "Impacts of MicroRNA-483 on Human Diseases"

_ncrna, 2023, doi:10.3390/ncrna9040037_

Round 1

Reviewer 1 Report

The manuscript titled “Impact of microRNA-483 on diabetes pathogenesis and complications” by Matson et al., explains the role of microRNA-483-5p and microRNA-483-3p in various biological functions and their associated diseases. The manuscript is well written by authors. There are a few major points that authors need to take care of before it gets accepted. Overall, I recommend this manuscript to get accepted after revision of few major points.

·       The manuscript discussed different diseases along with diabetes. However, authors need to check the title. It represents only diabetes. Authors need to pay attention to this.

·       Authors need to explain how microRNA-483-5p and microRNA-483-3p are correlated? Are these antagonists to each other? This point is confusing throughout the manuscript.

·       Most of the target genes for microRNA-483-5p and microRNA-483-3p are the same. How exactly are they involved in expression (up or down) pattern?  

·       As this is a review, it is better that authors need to make figures to represent (pictorial representation) the role of microRNA-483-5p and microRNA-483-3p in different biological functions/diseases. Reading this manuscript without figures is difficult to understand.    

·       Authors added two tables in the manuscript, however, they haven’t included it in any of the section.

·       Conclusions section is a bit confusing, specifically about miR-483-5p. Authors need to make it short and effective.   

·       Table 2 caption, “Implication”, ‘s’ needs to be added. No italics. Full stop after Table 2.

The manuscript is written well, however, authors need to check the typos.

Table 2 caption, “Implication”, ‘s’ needs to be added. No italics. Full stop after Table 2. 

Page 8, line 338. Space needs to check.

Reviewer 2 Report

This review comprehensively explores the role of miR-483-5p and miR-483-3p in type 2 diabetes and its related complications. It highlights that miR-483-5p serves as a protective factor in various insulin-sensitive tissues and a potential biomarker for metabolic risk, while miR-483-3p, often elevated in diabetes, can induce cell apoptosis and lipotoxicity. Overall this paper is well written. Here are several suggestions for the paper:

1. The apparent protective role of miR-483-5p in most metabolic diseases versus its elevated expression in cardiovascular disease presents a discrepency. The authors should propose a hypothesis to reconcile these contrasting observations. Furthermore, considering the common co-occurrence of diabetes and cardiovascular disease, any studies examining miR-483-5p expression in patients with both conditions should also be summarized and mentioned in the paper.

2. The authors should include a discussion on the challenges within this field of study and discuss any limitations present in the referenced articles

Author Response

Dear Editors,

We thank the Reviewers for their time and constructive comments. We have thoroughly revised the manuscript based on their suggestions. Please find below our point-to-point responses to the Reviewers’ comments.

Review 2:

Open Review

Comments and Suggestions for Authors

This review comprehensively explores the role of miR-483-5p and miR-483-3p in type 2 diabetes and its related complications. It highlights that miR-483-5p serves as a protective factor in various insulin-sensitive tissues and a potential biomarker for metabolic risk, while miR-483-3p, often elevated in diabetes, can induce cell apoptosis and lipotoxicity. Overall, this paper is well written. Here are several suggestions for the paper:

  1. The apparent protective role of miR-483-5p in most metabolic diseases versus its elevated expression in cardiovascular disease presents a discrepancy. The authors should propose a hypothesis to reconcile these contrasting observations. Furthermore, considering the common co-occurrence of diabetes and cardiovascular disease, any studies examining miR-483-5p expression in patients with both conditions should also be summarized and mentioned in the paper.

Answer: Thank you for your valuable input. We have addressed this point as suggested in section 4.3 (lines 292-299).

  1. The authors should include a discussion on the challenges within this field of study and discuss any limitations present in the referenced articles.

Answer: Thank you for pointing this out. We have discussed this as suggested in section 6.

Round 2

Reviewer 1 Report

The manuscript titled “Impacts of microRNA-483 on human diseases” by Matson et al., explains the role of microRNA-483-5p and microRNA-483-3p in various biological functions with associated diseases. Authors addressed all questions and changed the manuscript accordingly. I recommend accepting this manuscript.